# The dynamic adsorption affinity of ligands is a surrogate for the passivation of surface defects

Jian Xu [1,5], Aidan Maxwell[1,5], Zhaoning Song [2], Abdulaziz S. R. Bati [3], Hao Chen [1], Chongwen Li[1], So Min Park [1], Yanfa Yan [2], Bin Chen [1,3] & Edward H. Sargent [1,3,4]

Surface defects in semiconducting materials, though they have been widely studied, remain a prominent source of loss in optoelectronic devices; here we sought a new angle of approach, looking into the dynamic roles played by surface defects under atmospheric stressors and their chemical passivants in the lifetime of optoelectronic materials. We find that surface defects possess properties distinct from those of bulk defects. ab initio molecular dynamics simulations reveal a previously overlooked reversible degradation mechanism mediated by hydrogen vacancies. We find that dynamic surface adsorption affinity (DAA) relative to surface treatment ligands is a surrogate for passivation efficacy, a more strongly-correlated feature than is the static binding strength emphasized in prior reports. This guides us to design targeted passivator ligands with high molecular polarity: for example, 4-aminobutylphosphonic acid exhibits strong DAA and provides defect passivation applicable to a range of perovskite compositions, including suppressed hydrogen vacancy formation, enhanced photovoltaic performances and operational stability in perovskite solar cells.

Defects in semiconducting materials, present at increased densities at surfaces and interfaces compared to in the bulk, are a source of performance loss in optoelectronic devices[1,2]. While surface defects have been studied extensively from a static chemical and physical standpoint, it remains challenging to elucidate specifically the dynamic roles they play in accelerating degradation and worsening energetic losses under operating and atmospheric stressors[3,4].

To develop targeted surface engineering approaches towards high performance in optoelectronic devices, the following questions must first be tackled: (i) What are the dominant defects at the surface, and do these surface defects exhibit distinct properties compared to bulk defects? (ii) What is the dynamic behavior of surface defects

under device operating conditions such as exposure to oxygen, moisture and heat? (iii) What are the key microscopic mechanisms underlying the degradation processes?

In selecting a semiconductor material system to study computationally and experimentally, we decided to focus our investigations on mixed lead-tin (Pb-Sn) perovskites. Pb-Sn perovskites exhibit a high degree of sensitivity to surface defects, especially under atmospheric stressors such as oxidation; but are relevant to perovskite multi-junction photovoltaics[5–8] Prior theoretical calculations have indicated that bulk defects in Pb-Sn perovskites (50/50 Pb/Sn mixture) are relatively benign and do not create deep traps within the bandgap[9,10]; however, experimental studies have shown that surface defect

[1]Department of Electrical and Computer Engineering, University of Toronto, 35 St George Street, Toronto, ON M5S 1A4, Canada. [2]Department of Physics and Astronomy, and Wright Center for Photovoltaics Innovation and Commercialization, University of Toledo, 2801 W. Bancroft Street, Toledo, OH 43606, USA. [3]Department of Chemistry, Northwestern University, 2145 Sheridan Rd, Evanston, IL 60208, USA. [4]Department of Electrical and Computer Engineering, Northwestern University, 2145 Sheridan Rd, Evanston, IL 60208, USA. [5]These authors contributed equally: Jian Xu, Aidan Maxwell. ✉e-mail: bin.chen@northwestern.edu; ted.sargent@utoronto.ca

passivation is necessary to achieve both high efficiency and stability in Pb-Sn perovskite solar cells (PSCs)[5,11,12].

We reasoned that improving the efficiency and stability of mixed Pb-Sn PSCs will require a thorough examination of surface defect chemistry and physics; this understanding can then be levered for the rational molecular design of defect-passivating ligands. Prior studies have designed passivating molecules based on a static picture of the interaction between passivating ligands and perovskite surfaces under vacuum at 0 K[13,14]; these models do not take into account operational and atmospheric stressors.

Here we used ab initio molecular dynamics (AIMD) simulations to investigate the dynamic behavior of perovskite surface defects in the presence of atmospheric stressors including heat, oxygen, and moisture. With this approach, we obtain a clearer picture of defect behavior in response to various environmental stressors, and find a hydrogen vacancy-mediated degradation mechanism in mixed Pb-Sn perovskites. We observe that while certain passivators have a comparable static binding energy with the perovskite surface, these exhibit distinct dynamic adsorption abilities under operational conditions. We use this descriptor, dynamic adsorption affinity (DAA), as a guide in the design of passivators. Seeking to test these computational findings experimentally, we characterized 4-Aminobutylphosphonic acid (4-ABPA) as a passivator for mixed Pb-Sn PSC: the resultant Pb-Sn materials exhibit reduced non-radiative recombination and extended carrier lifetimes, with simultaneously suppressed degradation and improved operating stability in ambient conditions. We also show that this strategy is applicable to pure Pb perovskite compositions.

## Results and discussion
### Surface defect computational studies
We carried out defect calculations using density functional theory (DFT) with the Heyd–Scuseria–Ernzerhof (HSE) hybrid functional and including spin-orbit coupling (SOC). We identified the stable chemical potential regions in $MAPb_{0.5}Sn_{0.5}I_3$ (Fig. 1a) by considering several secondary phases (MAI, $SnI_2$, $PbI_2$, $SnI_4$, $MA_2SnI_6$, $MAPbI_3$ and $MASnI_3$). We also employed a recently-developed self-consistent potential correction (SCPC) method[15] for charged defect calculations in slabs. We found that neglecting corrections such as SCPC will produce unreliable (diverged) results (Fig.1b). Additional discussions on the calculations of charged surface defects can be found in Supplementary Note 1.

We first chose the thermodynamically more stable MAI-terminated (001) surface as the representative surface for investigation[3]. Surface defects exhibit distinct properties compared to bulk defects in mixed Pb-Sn perovskites (Fig.1c and Supplementary Fig. 1). Specifically, we observed that defects at the perovskite surface generally have lower defect formation energies ($E^f$) than those in the bulk, which is consistent with experimental results showing that grain surfaces have up to several orders of magnitude higher trap density than the bulk perovskite[16]. At the MAI-terminated perovskite surface, the dominant defect species which have a lower $E^f$ include MA vacancies ($V_{MA}$) and iodine vacancies ($V_I$). We found that, at the point B chemical potential condition and the pinned Fermi level (pinned $E_{FL}$, which is around the crossing point of the lowest-energy donor-like and acceptor-like defects[17]), the calculated $E^f$ of $V_{MA}$ in the bulk is 0.9 eV (Supplementary Fig. 1, middle panel), while its value at the perovskite surface is reduced to 0.4 eV (Fig. 1c, middle panel). This corresponds to a defect concentration that is (at room temperature) eight orders of magnitude higher at the surface compared to in the bulk, i.e. by a factor $e^{-\Delta E^f/k_B T}$ where $k_B$ is the Boltzmann constant and $T$ is the temperature.

We also observed that the donor-like defects tend to be deeper at the MAI-terminated perovskite surface than those in the bulk. In the bulk, $V_I$ defects are shallow with their charge-state transition levels (CTL) located close to the conduction band minimum (CBM)[9]. However, $V_I$ defects on the subsurface (L1 layer, Supplementary Fig. 2)

became deep with (0/1+) and (0/1−) CTLs located at 0.38 eV and 0.32 eV below the CBM, acting as potential non-radiative recombination centers. Similarly, Pb interstitials and Sn interstitials as donor defects also transform into deep defects at the perovskite surface. In the case of the MAI-terminated perovskite surface, this exhibits a more pronounced $p$-type feature than the bulk, as evidenced by its closer pinned $E_{FL}$ to the valence band maximum (VBM). This indicates severe self $p$-doping properties at the mixed Pb-Sn perovskite surface, a phenomenon exacerbated by the oxidation of $Sn^{2+}$ to $Sn^{4+}$.

We also considered the $PbI_2$-terminated surface which may be formed during solution-based post-treatment in experiments[14]. We found (Fig. 1d) that at the $PbI_2$-terminated surface, the dominant defects species include $V_I$ at the outermost surface (L0) layer, Sn vacancies ($V_{Sn}$) and iodine interstitials ($I_{interstitial}$). The acceptor-like defects ($V_{Sn}$, $V_{Pb}$ and $I_{interstitial}$) at the $PbI_2$-terminated surface tend to be deeper than those in the bulk and at the MAI-terminated surface. Specifically, $V_{Sn}$ exhibited a deep (0/2+) CTL located at 0.24 eV above the VBM. This we attributed to the lower VBM positions of the $PbI_2$-terminated surface compared to the MAI-terminated surface. The $PbI_2$-terminated surface is more $n$-type than the MAI-terminated surface (Supplementary Table 1), which is in agreement with previous experimentally-measured energetic levels of perovskite films obtained using ultraviolet photoelectron spectroscopy (UPS)[17].

### Degradation mechanisms in mixed Pb-Sn perovskites
We investigated the dynamic behavior of surface defects under device operating conditions such as exposure to oxygen, moisture and heat. We began with an AIMD simulation at room temperature (300 K) in the presence of oxygen, and we found that after reaching equilibrium, O-O and Sn-I bonds were broken, resulting in the formation of $V_{Sn}$ at the perovskite surface (Fig. 2a). These phenomena were not observed in static first-principles calculations that excluded temperature effects (Supplementary Fig. 3). This suggests the benefit that could be had in future studies wherein AIMD simulations are used to investigate temperature-dependent and time-dependent degradation process of perovskite materials; particularly for perovskite compositions which are highly sensitive to environmental conditions, such as mixed Pb-Sn perovskites.

We then investigated the dynamic behavior of surface defects under moisture and 300 K conditions. We observed that FA, MA and I escape from the surface over time, leaving vacant sites for water molecules to enter into the perovskite lattice (Fig. 2b and Supplementary Fig. 4). These vacancies will ultimately cause the material to decompose. Though A site vacancies and X site vacancies in the outermost surface layer (L0 layer, Supplementary Fig. 2) result in shallow traps near the band edges, they will still play an important role in limiting the durability of perovskite materials and acting as nucleation sites for degradation[18].

To understand the dynamic behavior of surface defects, we took simultaneously into account water, oxygen and temperature. We observed that at room temperature (Fig. 2c), one oxygen atom initially drags a hydrogen atom from the $MA^+$ molecule, leaving behind a hydrogen vacancy ($V_H(N)$ in $MA^+$). This is followed by another oxygen atom dragging a hydrogen atom from the $FA^+$ molecule (Fig. 2d), also resulting in a hydrogen vacancy ($V_H(N)$ in $FA^+$). Interestingly, we found that the deprotonation/protonation process for a $FA^+$ molecule is reversible at room temperature (Supplementary Fig. 5). However, at the film formation temperature (400 K), the deprotonation process is energetically favorable (Supplementary Fig. 6). Zhang et al.[19] found that the hydrogen vacancy ($V_H(N)$) in the $FA^+$ molecule had a higher formation energy than that of the $MA^+$ molecule, which is consistent with our AIMD observations that $FA^+$ is more stable than $MA^+$ under exposure to oxygen and moisture.

To probe this hydrogen-triggered degradation mechanism experimentally, we performed temperature-programmed desorption

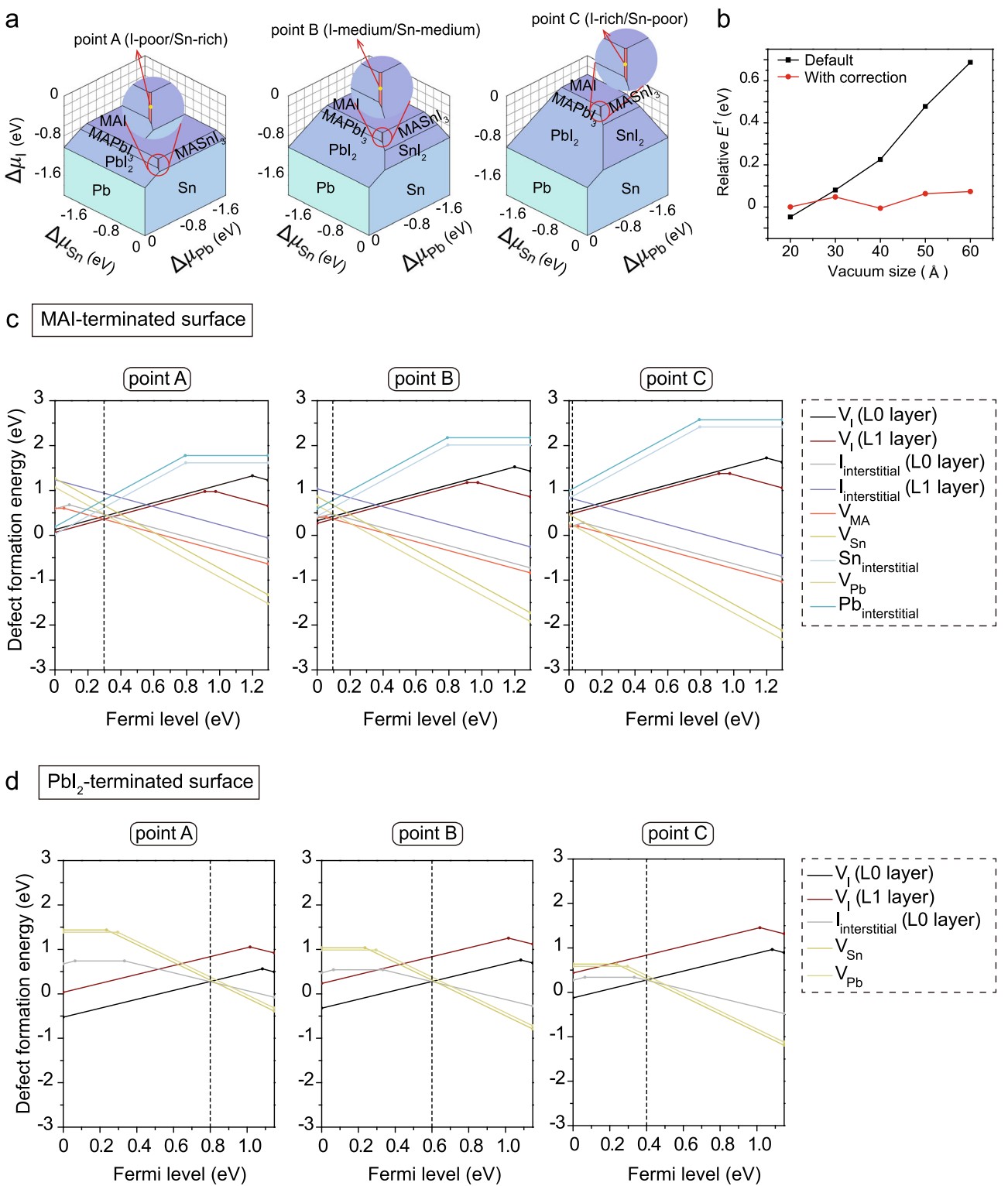

**Fig. 1 | Surface defect computational studies. a** Phase diagrams of MAPb$_{0.5}$Sn$_{0.5}$I$_3$ at the chemical potential of MA ($\Delta\mu_{MA}$) of −2.8 eV, −3.0 eV and −3.2 eV. Three chemical potential points on the equilibrium region have been highlighted as yellow dots: point A (I-poor/Sn-rich), point B (I-medium/Sn-medium) and point C (I-rich/Sn-poor). **b** Perdew-Burke-Ernzerhof functional (PBE) calculated relative defect formation energies ($E^f$) of the negatively charged MA vacancy ($V_{MA}$) as a function of vacuum size with and without self-consistent potential correction (SCPC). Heyd−Scuseria−Ernzerhof (HSE) including spin-orbital coupling (SOC) calculated $E^f$ of native defects at the **c** MAI-terminated (001) surface and **d** PbI$_2$-terminated (001) surface of MAPb$_{0.5}$Sn$_{0.5}$I$_3$ under the chemical potential conditions of points A, B and C. The black dashed line indicates the crossing point of the lowest-energy donor-like and acceptor-like defects. $V_I$ iodine vacancy, $V_{MA}$ MA vacancy, $V_{Sn}$ Sn vacancy, $V_{Pb}$ Pb vacancy, $I_{interstitial}$ iodine interstitial, $Sn_{interstitial}$ Sn interstitial, $Pb_{interstitial}$ Pb interstitial.

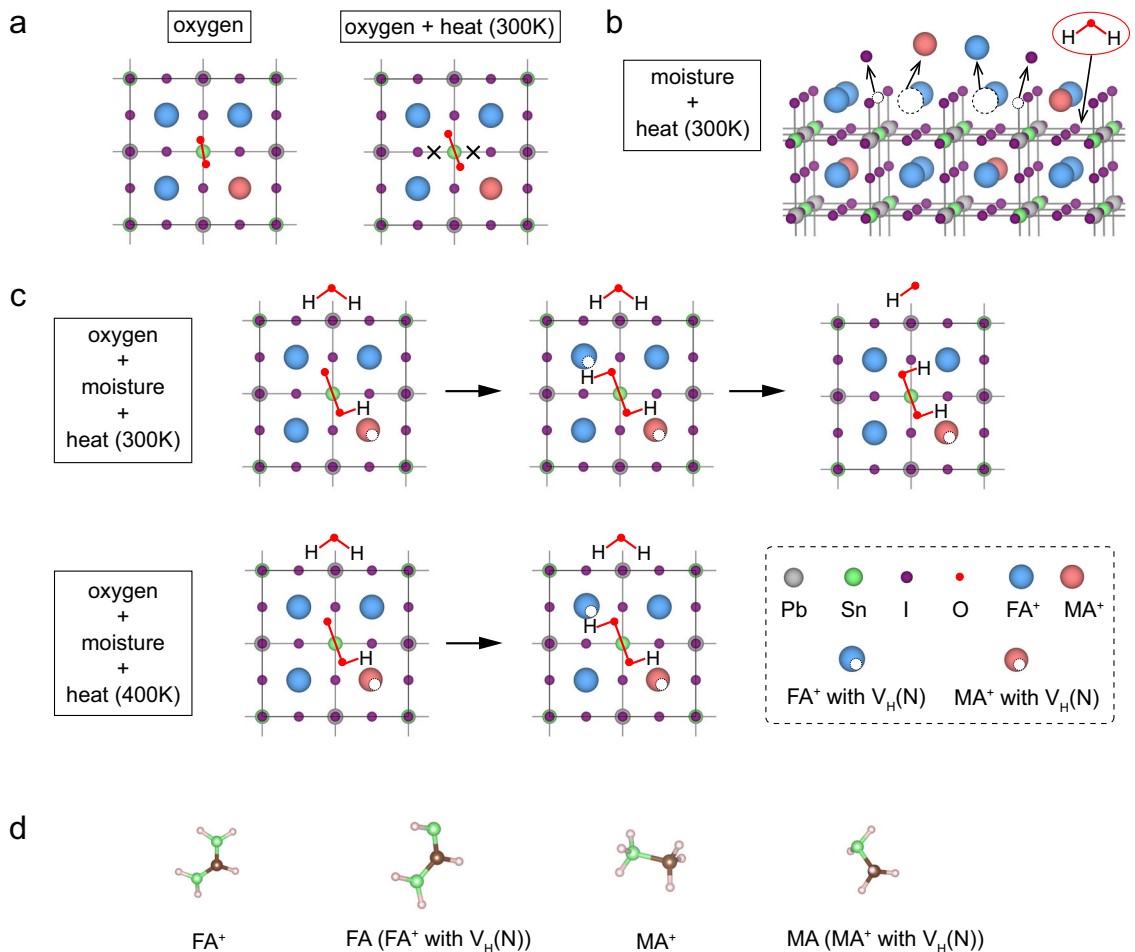

**Fig. 2 | Exploration of candidate degradation mechanisms in mixed Pb-Sn perovskite.** Schematic diagrams of ab initio molecular dynamics (AIMD) simulation for $MA_{0.25}FA_{0.75}Pb_{0.5}Sn_{0.5}I_3$ (001) surface **a** exposure to oxygen conditions, **b** exposure to moisture conditions, and **c** exposure to both oxygen and moisture conditions at room temperature (300 K) and film formation temperature (400 K). **d** Molecular structures of $FA^+/MA^+$ and FA/MA. N atoms, green color; C atoms, brown color; H atoms, light pink color.

mass spectrometry (TPD-MS) measurements of mixed Pb-Sn perovskite to track the volatile species released from the perovskite surface, here focusing in on the device-relevant composition of $MA_{0.25}FA_{0.75}Pb_{0.5}Sn_{0.5}I_3$. Prior to the TPD-MS measurement, the perovskite film was exposed to humid ambient air to trigger surface degradation. We found (Fig. 3e) that the perovskite film released $V_H(N)$-contained MA ($CH_3NH_2$) and hydriodic acid (HI) gas species at a low temperature of ~60 °C; an effect which became more pronounced with increasing temperature. However, we observed the decomposition of $FA^+$ into $V_H(N)$-contained FA ($H_2N$-CH=NH) at a higher temperature of ~80 °C, which indicates that increased FA content is beneficial for the improved thermal stability of mixed Pb-Sn perovskite films. The creation of the hydrogen vacancies in the $MA^+/FA^+$ acts as a source of instability, making the molecules more likely to be expelled from the surface of the perovskite, enabling further $O_2$ and $H_2O$ penetration into the lattice.

### Rational design of passivators based on DAA

In light of the above findings, we posited that surface passivators for mixed Pb-Sn perovskites should ideally have multi-functional units that incorporate both electron-rich and electron-poor domains (Fig. 3a). We reasoned (Supplementary Fig. 7) that a higher maximum electrostatic potential ($\varphi_{max}$) at the electron-poor side ($NH_3^+/NH_2$) could enhance the binding strength between the passivators and the acceptor-like defects ($V_A$, $V_{Sn}$) at the perovskite surface;

simultaneously, a lower minimum electrostatic potential ($\varphi_{min}$) at the electron-rich side ($SO_3/SO_2/PHO_3$) could increase the binding strength between the passivators and the donor-like defects ($V_I$, under-coordinated $Pb^{2+}/Sn^{2+}$) at the perovskite surface. The interaction mechanism of this type of ligands with the perovskite surface is discussed in Supplementary Note 2 and Supplementary Fig. 8. A larger difference between $\varphi_{max}$ and $\varphi_{min}$ within a ligand usually indicates a higher molecular polarity. With this in mind, we designed two passivators – 3-amino-1-propanesulfonic acid (APSA) and 4-aminobutylphosphonic acid (4-ABPA) – with higher molecular polarity (Fig. 3a) compared to formamidine sulfinic acid (FSA), which is a zwitterionic antioxidant had been successfully utilized for surface passivation in mixed Pb-Sn perovskites[12].

We conducted AIMD simulations to investigate the dynamic adsorption capacity of APSA and 4-ABPA on the perovskite surface under oxygen and moisture exposure at a temperature of 400 K (Fig. 3c). Figure 3d shows the fluctuation in the number of adsorbed ligands during the 4 ps to 12 ps AIMD simulation. Our findings indicate that eleven 4-ABPA ligands were adsorbed on the perovskite surface through strong O-Pb (Sn) coordinate bonds. However, in the case of APSA adsorption, only 7 ~ 8 under-coordinated $Pb^{2+}/Sn^{2+}$ sites at the perovskite surface were coordinated. Moreover, during the last 2 ps AIMD simulation, the number of adsorbed ligands showed a wider fluctuation range (4 ~ 10) in the case of APSA passivation, while 4-ABPA has a narrow fluctuation range of 11 ~ 12, indicating more stable surface

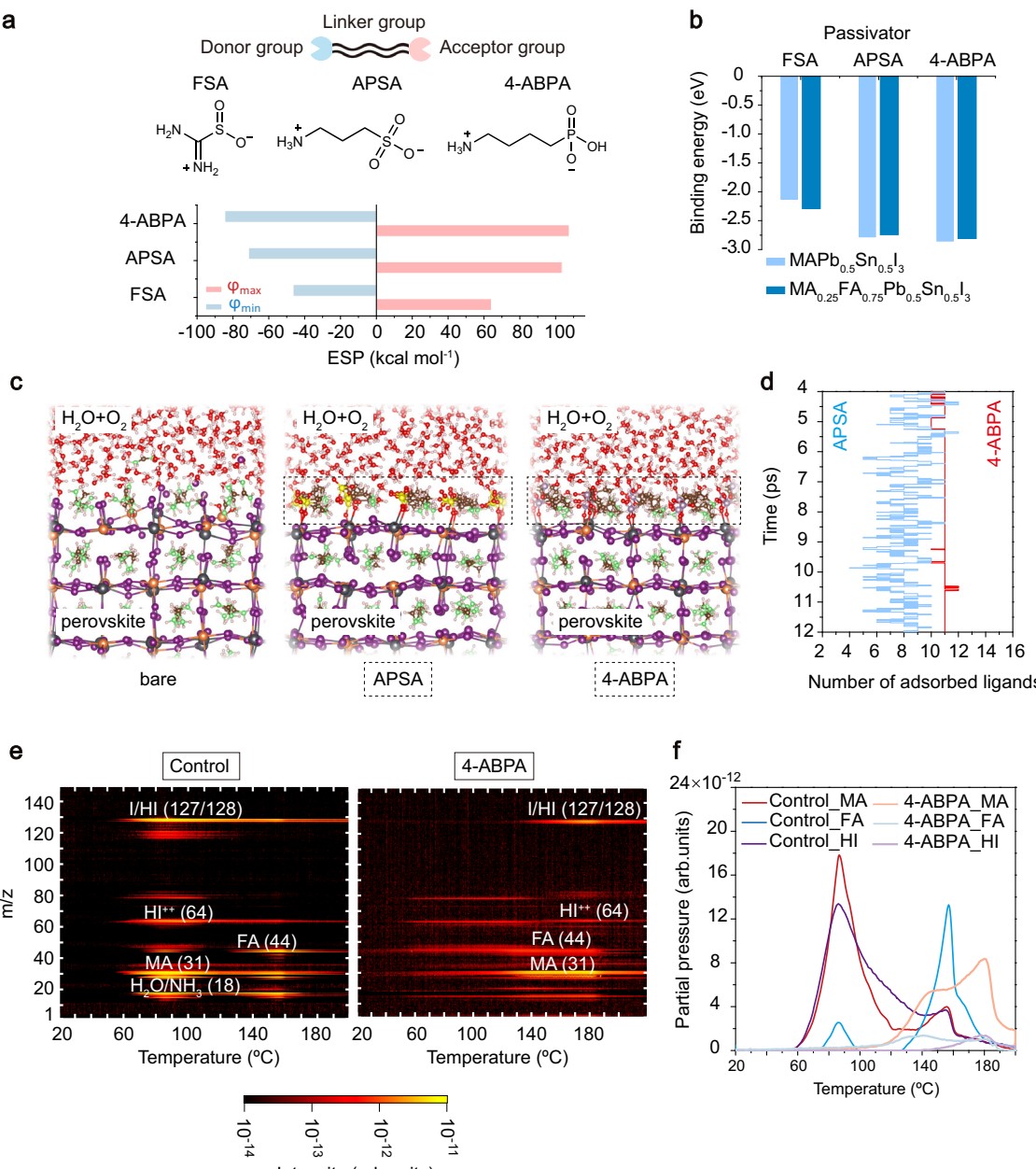

**Fig. 3 | Design of passivators. a** Molecular structures of three passivator molecules (FSA, APSA and 4-ABPA). Gaussian calculated electrostatic potentials ($\varphi$, ESP) values of maximum ESP ($\varphi_{max}$) and minimum ESP ($\varphi_{min}$) are also shown. $\varphi_{max}$ and $\varphi_{min}$ values are obtained using Multiwfn code[38]. **b** Binding energy of three molecules with the Al-terminated (001) surface of $MAPb_{0.5}Sn_{0.5}I_3$ and $MA_{0.25}FA_{0.75}Pb_{0.5}Sn_{0.5}I_3$. **c** ab initio molecular dynamics (AIMD) snapshots of bare, APSA, and 4-ABPA adsorbed perovskite surface exposure to oxygen and moisture conditions at a temperature of 400 K. The perovskite composition is $MA_{0.25}FA_{0.75}Pb_{0.5}Sn_{0.5}I_3$. P atoms, lavender color; S atoms, yellow color; O atoms, red color; Pb atoms, gray color; Sn atoms, orange color; I atoms, purple color; N atoms, green color. **d** The number of adsorbed ligands for APSA and 4-ABPA at the conditions shown in Fig. 3c from 4 ps to 12 ps during the AIMD simulation. Ligand number of 16 represents complete surface coverage within the simulation unit. Blue line: APSA adsorption case; Red line: 4-ABPA adsorption case. The average value of adsorbed ligands during the last 2 ps of the AIMD simulation for APSA and 4-ABPA are 8 and 11, respectively. **e** Temperature-programmed desorption mass spectrometry (TPD-MS) of control and 4-ABPA treated $FA_{0.75}MA_{0.25}Pb_{0.5}Sn_{0.5}I_3$ perovskite films. The control film exhibits a decomposition onset temperature at 60 °C and first peak at ~85 °C whereas the 4-ABPA treated film shows a higher decomposition onset at ~120 °C and peaks at >150 °C (Fig. 3f). **f** Characteristic mass spectrum ($m/z$) peaks of control and 4-ABPA treated perovskite films.

binding. Though APSA and 4-ABPA have a comparable static binding energy with the perovskite surface (Fig. 3b), the latter exhibited a much stronger DAA and improved stability under operational conditions. We attribute this to the better defect passivation effect of 4-ABPA enabled by its increased molecular polarity and the increased hydrophobicity of $PHO_3^-$ functional group.

To test these findings experimentally, we conducted TPD-MS measurements on untreated (control) and 4-ABPA passivated

perovskite films. The control film exhibits a decomposition onset temperature at 60 °C and first peak at ~85 °C whereas the 4-ABPA treated film shows a higher decomposition onset at ~120 °C and peaks at >150 °C (Fig. 3f). The results indicate suppressed surface degradation in the 4-ABPA passivated film compared to the control film after exposure to humid air. Importantly, we found that the partial vapor pressure of released HI gas was reduced by more than one order of magnitude in the 4-ABPA passivated films compared to the control film

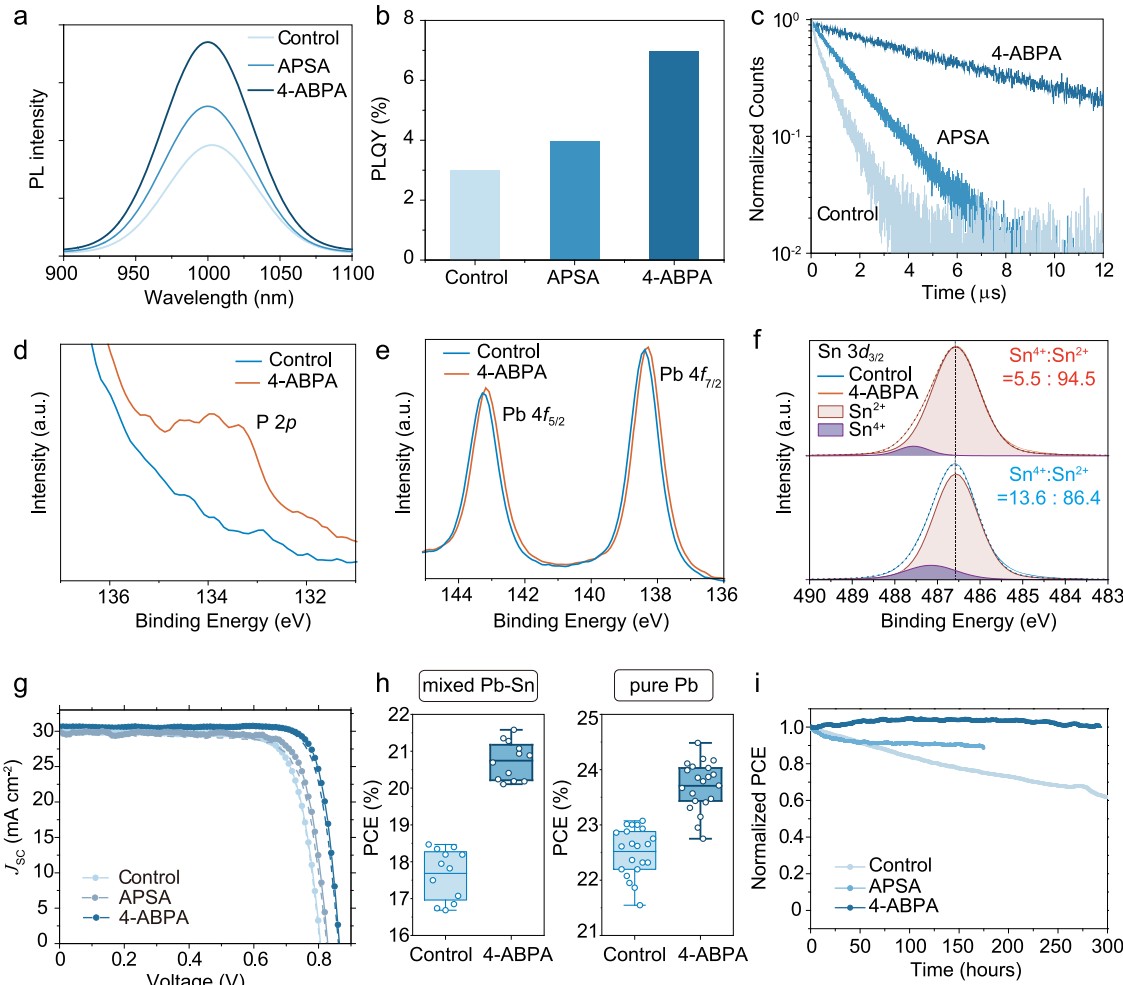

**Fig. 4 | Optoelectronic and photovoltaic properties in experiments.**
**a** Photoluminescence (PL) spectra, **b** photoluminescence quantum yield (PLQY) results, and **c** time-resolved photoluminescence (TRPL) spectra for control, APSA and 4-ABPA-treated Pb-Sn perovskite films. X-ray photoelectron spectroscopy (XPS) spectra of **d** P 2*p*, **e** Pb 4*f* and **f** Sn 3*d* for control and 4-ABPA- treated Pb-Sn perovskite films. a.u.: arbitrary units. **g** Champion forward (solid) and reverse (dashed) *J-V* curves of control, APSA and 4-ABPA-treated Pb-Sn perovskite solar cells (PSCs). $J_{SC}$: short-circuit current density. **h** Device performance statistics of Pb-

Sn and pure-Pb PSCs. PCE: power conversion efficiency. The box plot denotes minima (bottom line), maxima (top line), median (center line), 75th (top edge of the box), 25th (bottom edge of the box) percentiles. The dots represent each device datum. The sample sizes for both the control and 4-ABPA-treated Pb-Sn devices are 12, while the sample sizes for the control and 4-ABPA-treated pure-Pb devices are both 21. **i** Maximum power point (MPP) stability tracking of control, APSA and 4-ABPA treated Pb-Sn PSCs in ambient conditions.

(Fig. 3f). Additionally, the release of FA and MA was significantly reduced in the 4-ABPA passivated film. This decrease in partial vapor pressures of these volatile decomposition products in the 4-ABPA passivated film indicates a suppression of hydrogen vacancies in the $MA^+/FA^+$ species, retarding the thermal decomposition of Sn-Pb perovskites, and improving surface stability.

## Optoelectronic and photovoltaic properties

Encouraged by the theoretical surface-passivating abilities of APSA and 4-ABPA, we sought to employ these ligands experimentally via solution-based surface passivation of mixed Pb-Sn perovskites with a composition of $MA_{0.25}FA_{0.75}Pb_{0.5}Sn_{0.5}I_3$. To probe the efficacy of each ligand, we carried out photoluminescence (PL) and photo-luminescence quantum yield (PLQY) measurements of control (untreated), APSA-treated and 4-ABPA-treated perovskite thin films on quartz substrates. PL spectra (Fig. 4a) indicate stronger luminescence after surface treatment, with 4-ABPA showing the most significant improvement. This result is corroborated by PLQY results (Fig. 4b), in which surface treatment with 4-ABPA resulted in a 2.3-fold

enhancement from 3.0% for control films to 7.0%. These results are consistent with the expected reduction in non-radiative recombina-tion associated with surface defect passivation predicted by our cal-culations. To probe the effect of surface treatment on the carrier lifetimes of the Pb-Sn films, we carried out time-resolved photo-luminescence (TRPL) measurements (Fig. 4c). Weighted average car-rier lifetimes <τ> using a biexponential fitting model were calculated to be 0.52, 1.29 and 6.49 μs for control, and APSA and 4-ABPA-treated films, respectively.

We selected 4-ABPA as the most promising candidate for appli-cation in devices based on our theoretical findings and steady-state and time-resolved PL results. To probe the surface binding properties of 4-ABPA experimentally, we then turned to X-ray photoelectron spectroscopy (XPS) of control and 4-ABPA-treated perovskite films exposed to air for 1 min (Fig. 4d–f and Supplementary Fig. 9). We observed the emergence of a peak at ~133.5 eV in the shoulder of the Pb $4f_{7/2}$ peak which we attribute to P 2*p* from the phosphonic acid func-tional group of 4-ABPA (Fig. 4d). In addition, a slight shift to a lower binding energy was observed for the Pb 4*f*, I 3*d* and N1*s* peaks,

**Table 1 | Photovoltaic performance of champion control, APSA and 4-ABPA-treated Pb-Sn perovskite solar cells**

|  | Control | APSA treated | 4-ABPA treated |
|---|---|---|---|
| PCE (%) | 18.4 | 19.8 | 21.6 |
| $V_{OC}$ (V) | 0.80 | 0.83 | 0.86 |
| $J_{SC}$ (mA cm$^{-2}$) | 30.2 | 30.0 | 30.6 |
| FF (%) | 76.3 | 79.6 | 81.9 |

*PCE* power conversion efficiency, *$V_{OC}$* open-circuit voltage, *$J_{SC}$* short-circuit current density, FF fill factor.

consistent with an increase in the local electron density at the perovskite surface (Fig. 4e and Supplementary Fig. 9). Furthermore, the $Sn^{4+}$ content was reduced from 13.6% for control films to 5.5% for 4-ABPA-treated films (Fig. 4f), indicative of suppressed Sn oxidation. We also note a reduction in the intensity of the $O_I$ peak at ~530.2 eV, commonly attributed to metal-oxides (i.e. Sn-O or Pb-O in our case), after 4-ABPA treatment (Supplementary Fig. 9).

Next, we fabricated p-i-n Pb-Sn perovskite solar cells with a device structure of ITO/PEDOT:PSS/Pb-Sn perovskite/C$_{60}$/BCP/Ag. 4-ABPA-treated devices achieved an open-circuit voltage ($V_{OC}$) and power conversion efficiency (PCE) of 0.86 V and 21.6% respectively, representing a significant improvement in comparison to control devices, which achieved 0.80 V and 18.4%, respectively (Fig. 4g and Table 1). External quantum efficiency (EQE) measurements also revealed a slight improvement in short-circuit current density ($J_{SC}$) from 29.8 to 30.2 mA cm$^{-2}$, consistent with a higher-quality perovskite/C$_{60}$ interface and reduced surface Sn oxidation (Supplementary Fig. 10). Device performance statistics are shown in Supplementary Fig. 11. We also fabricated APSA-treated Pb-Sn devices, which delivered a $V_{OC}$ and PCE of 0.83 V and 19.8%, respectively (Table 1). The relative increase in $V_{OC}$ is in good agreement with the trend observed in the PLQY measurements (Fig. 4b) and the calculated quasi-Fermi level splitting (QFLS) values (Supplementary Note 3 and Supplementary Fig. 12).

We also demonstrate the applicability of our strategy beyond Pb-Sn perovskites, achieving a champion PCE of 24.4% for pure Pb perovskite (-1.55 eV) solar cells after 4-ABPA treatment, compared to 23.3% for control devices (Fig. 4h, discussions in Supplementary Note 4, Supplementary Figs. 13–16).

Finally, we carried out maximum power point (MPP) tracking of encapsulated control, APSA and 4-ABPA treated Pb-Sn devices under continuous illumination and ambient conditions (Fig. 4i). The control device delivered a $T_{80}$ (the time taken for device efficiency to drop to 80% of its initial value) of 130 h. In comparison, the APSA treated device was reduced to 90% of its initial PCE after 170 hours, while the 4-ABPA treated Pb-Sn devices show little degradation after 290 h. These findings suggest that the 4-ABPA treatment substantially enhances device stability over the control and APSA treatments.

## Methods
### Computational details
We used the Vienna Ab initio Simulation Package (VASP)[20] to perform DFT-based first-principles calculations. For the exchange-correlation functional, the Perdew–Burke–Ernzerhof functional (PBE)[21] and the screened Heyd–Scuseria–Ernzerhof (HSE) hybrid functional[22,23] were adopted. In addition, the DFT-D3 method was used for the van der Waals (vdW) correction[24]. The spin-orbital coupling (SOC) effect was included. The mixing parameter (α) of the Hartree–Fock term in HSE + SOC calculations was set to 0.30 in MAPb$_{0.5}$Sn$_{0.5}$I$_3$, which has been proven to reproduce the experimental bandgap. The structure of MAPb$_{0.5}$Sn$_{0.5}$I$_3$ was provided in Supplementary Data 1. We used the cutoff energy of 400 eV with the energy and force convergence tolerance setting to 10$^{-5}$ eV and 0.02 eV·Å$^{-1}$, respectively.

### Defect thermodynamics
The defect formation energy can be calculated as[25–27]

$$E^f[X^q] = E_{tot}[X^q] - E_{tot}[clean] + \sum_i n_i\mu_i + q(\varepsilon_{VBM} + E_F) + E_{corr} \quad (1)$$

Where $E_{tot}[clean]$ and $E_{tot}[X^q]$ were the total energy of a supercell without and with defects, respectively. $\mu_i$ was the chemical potential of the component element. $\varepsilon_{VBM}$ and $E_F$ were the energy levels of the valence band maximum (VBM) and the Fermi level measured from the VBM, respectively. To account for the finite-size effects, several important correction schemes, e.g, the potential-alignment correction ($q\Delta V$), were considered in our defect calculations[25,26]. For charged defect calculations in surface slabs, we employed a recently developed SCPC method[15]. This method has been implemented into the official VASP package from version 6.2. onwards.

To obtain the defect formation energies ($E^f[X^q]$) in mixed Pb-Sn perovskites, we identified the stable chemical potential regions by considering several secondary phases (MAI, SnI$_2$, PbI$_2$, SnI$_4$, MA$_2$SnI$_6$, MAPbI$_3$ and MASnI$_3$) and selected three representative chemical potential conditions, e.g., point A (I-poor/Sn-rich), point B (I-medium/Sn-medium) and point C (I-rich/Sn-poor), as shown in Fig. 1a. The phase diagrams were visualized by the Chesta code.

The charge-state transition levels (CTLs) were defined as[28–30]

$$\varepsilon(q_1/q_2) = \frac{(E_{tot}[X^{q_1}] + E_{corr}[X^{q_1}]) - (E_{tot}[X^{q_2}] + E_{corr}[X^{q_2}])}{q_2 - q_1} - \varepsilon_{VBM} \quad (2)$$

These CTLs correspond to the the Fermi-level positions at which a defect changes its charge state from $q_1$ to $q_2$, i.e., defects in charge state $q_1$ and $q_2$ have equal formation energies.

### Ab-initio molecular dynamic (AIMD) simulations
AIMD simulations were performed with the CP2K package[31] in the constant-volume and constant-temperature (NVT) ensemble. The temperature was controlled with Nosé-Hoover thermostat[32] at room temperature (300 K) and 400 K. PBE-D3 functional was used with double-zeta basis sets (DZVP-MOLOPT)[33] and Goedecker-Teter-Hutter (GTH) pseudopotentials[34]. The cut-off was set to 560 Ry. We constructed the FA$_{0.75}$MA$_{0.25}$Pb$_{0.5}$Sn$_{0.5}$I$_3$ (001) surface model in AIMD simulations. We initially put 16 passivator molecules at the perovskite surface to represent complete surface coverage within the simulation unit (4×4 supercell). The H$_2$O layer in the FA$_{0.75}$MA$_{0.25}$Pb$_{0.5}$Sn$_{0.5}$I$_3$/H$_2$O interface model employed the experimental density of liquid water. We used the Packmol software[35] to build the starting structure of H$_2$O layer. The initial and final configurations of the AIMD simulations were provided in Supplementary Data 2. The AIMD simulations are run for 10 - 20 ps with the time step of 1.0 fs to ensure equilibrium. In our AIMD simulations (Fig. 3c, d), the passivator molecules were first added to a pristine (clean) perovskite surface, and then adsorption was studied under the environmental stressors.

### Materials
All materials were used as received without further purification. Commercial ITO substrates (20 Ω sq$^{-1}$) with 25 mm × 25 mm dimension were purchased from TFD Inc. The organic halide salts (FAI, MAI) as well as 4-Fluorophenethylammonium bromide (4F-PEABr) were purchased from GreatCell Solar Materials (Australia). Poly(3,4-ethylenedioxythiophene) polystyrene sulfonate (PEDOT: PSS) aqueous solution (Al 4083) was purchased from Heraeus Clevios (Germany). PbI$_2$ (99.99%), and CsI (99.99%) were purchased from TCI Chemicals. SnI$_2$ (99.99%), SnF$_2$ (99%), 3-Amino-1-propanesulfonic acid (APSA) and Guanidine thiocyanate (GuaSCN, 99%) were purchased from Sigma-Aldrich. C$_{60}$ and Bathocuproine (BCP) were purchased from Xi'an Polymer Light Technology (China). 4-Aminobutylphosphonic acid

(4-ABPA) was purchased from Ambeed. All the solvents used in the process were anhydrous and purchased from Sigma-Aldrich.

### Perovskite precursor solution

1.8 M narrow-bandgap perovskite precursor solution with a composition of $FA_{0.75}MA_{0.25}Pb_{0.5}Sn_{0.5}I_3$ was prepared by dissolving, FAI, MAI, $SnI_2$ and $PbI_2$ in the mixed solvents of DMF and DMSO with a volume ratio of 3:1. Tin powders (5 mg), and $SnF_2$ (14 mg) were added to the precursor solution. For solar cell device fabrication, GuaSCN (4 mg), and 4F-PEABr (2 mg) were also added to the precursor solution. The precursor solution was then stirred at room temperature for 1 h. The precursor solution was filtered using a 0.22 μm Polytetra-fluoroethylene (PTFE) membrane before using.

### Pb-Sn perovskite film fabrication

Quartz substrates were sequentially cleaned using acetone and isopropanol, followed by UV-Ozone treatment for 15 min. The perovskite films were deposited with a two-step spin-coating procedure: (1) 1000 rpm for 10 s with an acceleration of 200 rpm s$^{-1}$, (2) 3800 rpm for 45 s with an acceleration of 1000 rpm s$^{-1}$. 300 μl CB was dropped onto the spinning substrate during the second spin-coating step at 18 s before the end of the procedure. The substrates were then treated on hotplate for 10 min at 100 °C. Post-treatment with APSA and 4-ABPA were carried out by spin-coating 1.5 mM solutions of each passivator (stirred overnight and filtered before use) in 1:1 IPA:Toluene at 4000 rpm for 25 s, followed by annealing at 100 °C for 5 min.

### Pb-Sn perovskite solar cell fabrication

The prepatterned indium tin oxide (ITO) glass substrates were sequentially cleaned using acetone and isopropanol. PEDOT: PSS (1:2 with IPA) was spin-coated on ITO substrates at 4000 rpm for 30 s and annealed on a hotplate at 150 °C for 20 min in ambient air. After cooling, we transferred the substrates immediately to a nitrogen-filled glovebox for the deposition of perovskite films. The perovskite films deposition and 4-ABPA post-treatment were carried out as described above. 25 nm C60, 8 nm BCP and 140 nm Ag were sequentially deposited on top of the perovskite layer by thermal evaporation. Experimental details of pure-Pb perovskite solar cell fabrication can be found at Supplementary Method.

### Device testing

The current density-voltage (J-V) characteristics were measured using a Keithley 2400 source meter under illumination from a solar simulator (Newport, Class A) with a light intensity of 100 mW cm$^{-2}$ (checked with a calibrated reference solar cell from Newport). J-V curves were measured in a nitrogen atmosphere with a scanning rate of 100 mVs$^{-1}$ (voltage step of 10 mV and delay time of 200 ms). The active area was determined by the aperture shade mask (0.049 cm$^2$ for small-area devices) placed in front of the solar cell. A spectral mismatch factor of 1 was used for all J-V measurements. For stabilized output measurements at MPP, the device testing chamber was left under ambient conditions. Solar cells were fixed at the MPP voltage, (determined from J-V sweeps in both scanning directions) and current output was tracked over time. EQE measurements were performed in ambient air using a QuantX-300 Quantum Efficiency Measurement System (Newport) with monochromatic light focused on the device pixel and a chopper frequency of 20 Hz.

### Stability testing

Devices were placed in a homemade stability tracking station. A white light-emitting diode (LED) light with an intensity of 100 mW cm$^{-2}$ was used. The spectrum of the LED simulator we used for MPP stability tracking was provided in our previous paper[36]. MPP were tracked by a perturb and observe algorithm that updates the MPP point every

20 min. Encapsulation was done by capping device with a glass slide, using UV-adhesive (Lumtec LT-U001) as sealant.

### PLQY measurements

The excitation source was an unfocused beam of a 405 nm c.w. diode laser. Photoluminescence was collected using an integrating sphere with a pre-calibrated fiber coupled to a spectrometer (Ocean Optics QE Pro) with an intensity of ~300 mW cm$^{-2}$. PLQY values were calculated by $PLQY = \frac{P_S}{P_{Ex} \cdot A}$, where $A = 1 - \frac{P_L}{P_{Ex}}$, $P_S$ is the integrated photon count of sample emission upon laser excitation; $P_{Ex}$ is the integrated photon count of the excitation laser when the sample is removed from integrating sphere, and $P_L$ is the integrated photon count of excitation laser when sample is mounted in the integrating sphere and hit by the beam. A set of neutral density filters were used to vary the excitation density.

### TRPL spectroscopy

TRPL measurements were performed using a Horiba Fluorolog Time Correlated Single Photon Counting (TCSPC) system with photo-multiplier tube detectors. A pulsed laser diode (504 nm, 110–140 ps pulse width, average 1.4 mW) was used as excitation sources for steady-state and transient measurements. TRPL decay curves were fitted with biexponential components to obtain a fast and a slow decay lifetime. The mean carrier lifetimes τ for the biexponential fit were calculated by the weighted average method.

### TPD-MS measurements

TPD-MS measurements were measured using a custom temperature-programmed desorption system reported previously[37]. The perovskite samples were shipped in air-tight containers and stored in a nitrogen glovebox as received. The samples were exposed to humid ambient air for ~5 h before the mass spectrometry experiment. A perovskite film deposited on an ITO glass substrate was cut into a 10 mm by 25 mm piece and placed in a quartz tube connected to the main chamber equipped with a quadrupole mass spectrometer (Stanford Research System, RGA 300). The system was pumped down overnight to allow degassing of the specimen before the thermal desorption/decomposition measurement. The evolution of the mass spectrum with a range of 1 to 200 atomic mass unit (AMU) was recorded in a 10-s interval. Prior to the heating, background signals were first collected for 10 min in the dark, and the averages were subtracted from the main signals. For the temperature-dependent desorption measurement, the sample was heated up from 20 to 200 °C at a ramp rate of 2 °C min$^{-1}$ and maintained at 200 °C using a tube furnace.

### XPS measurements

High-resolution XPS scans of C 1s, O 1s, N1s, P2p, Pb4f, and Sn 3d were performed with a monochromatic aluminum X-ray source (model 5600, PerkinElmer). The perovskite films were aged in ambient air for 1 min prior to measurement.

### Reporting summary

Further information on research design is available in the Nature Portfolio Reporting Summary linked to this article.

## Data availability

The main data supporting the findings of this study are available within the Article and its Supplementary Information. Source data are provided with this paper.

## Code availability

The VASP code for the numerical simulations in this work can be found at https://www.vasp.at. The CP2K code can be found at https://www.cp2k.org/. The Gaussian code can be found at https://gaussian.com/. The Chesta code can be found at https://n-hatada.github.io/chesta.

The Multiwfn code can be found at http://sobereva.com/multiwfn. The Packmol software can be found at https://m3g.github.io/packmol/.

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

## Acknowledgements

This research was made by King Abdullah University of Science and Technology (KAUST) Office of Sponsored Research (OSR) under award no. OSR-2020-CRG9-4350.2. This work was also supported by the U.S. Department of the Navy, Office of Naval Research (N00014-20-1-2572). SciNet is funded by the Canada Foundation for Innovation under the auspices of Compute Canada. The Work at the University of Toledo was supported by the U.S. Department of Energy's Office of Energy Efficiency and Renewable Energy (EERE) under the Solar Energy Technologies Office Award Number DE-EE0008753. A.S.R.B. acknowledges support from KAUST through the Ibn Rushd Postdoctoral Fellowship Award.

## Author contributions

J.X conceived the idea. J.X, A.M and E.H.S designed the project. J.X performed all DFT calculations, including defect calculations and AIMD simulations. A.M fabricated the mixed Pb-Sn PSCs and performed XPS and PL characterizations. Z.S and C.L carried out TPD-MS measurement under the supervision of Y.Y. A.S.R.B. performed MPP stability measurement. H.C fabricated the pure Pb PSCs. B.C and E.H.S supervised the work. J.X and A.M wrote the draft manuscript, and S.M, Y.Y, B.C and E.H.S improved the manuscript. All authors discussed the results and commented on the manuscript.

## Competing interests

The authors declare no competing interests.
