## [Peer Review File · Nature Communications]

The Dynamic Adsorption Affinity of Ligands is a Surrogate for the Passivation of Surface Defects

|REVIEWER COMMENTS

Reviewer #1 (Remarks to the Author):

In this study, the author introduced ab initio molecular dynamics (AIMD) simulation to reveal dynamic roles played by surface defects under atmospheric stressors and their chemical passivants. The author claimed the dynamic surface adsorption affinity (DAA) obtained by applying AIMD to perovskite is a surrogate for passivation efficacy. The ligand designed with consideration for the DAA (4-ABPA) improves the surface stability of perovskite. As a result, 4-ABPA treated perovskite enhanced a PCE of 21.6% and MPP stability over 240 hours under ambient conditions. This work involves applying interesting simulation techniques to confirm the surface stability of Sn incorporate perovskite film. Surface stability is very important in perovskite; therefore, this work is important and novel. However, to make the study more solid, the following comments should be supplemented before publication.

A few comments:

1. In page 4, author claims that 'At the MAI-terminated perovskite surface, the dominant defect species which have a lower E_f are MA vacancies (VMA), iodine vacancies (VI), and iodine interstitials (Iinterstitial) situated at the outermost surface (L0) layer...'. The author should clarify where they found the lowest E_f in Figure 1d. For example, VPb has the lowest defect formation energy in most areas. But, VPb is not the dominant defect species.
2. In page 5, author claims that 'the E_f of VMA in the bulk is 0.9 eV, while its value at the perovskite surface is reduced to 0.4 eV, causing a defect concentration that is eight orders of magnitude higher (at the surface compared to in the bulk) at room temperature. ...'. Is this finding a result of your work or previous research? The author should clarify the evidence for these statements.
3. In page 7, the authors argued that AIMD simulation at RT(300K) can confirm O-O and Sn-I bonds breaking and formation of VS_n at the surface. And this phenomenon were not observed in the static DFT calculation performed at 0K (Supplementary Fig 2). However, static DFT also can calculate the surface exposure at 300K. And in supplementary Fig 2b, we can find formation of VS_n in static DFT simulation at 300K. Author should specifically and clearly describe the advantages of AIMD simulation (300K) over static DFT (300K). This should be a key part of this research.
4. In this work, the author argues that DAA relative to surface treatment ligand is a surrogate for passivation efficacy and DAA is a more strongly correlated feature than the static binding strength. The authors compared APSA and 4-ABPA ligands with similar static binding energies and showed that the 4-ABPA ligand with higher DAA had a better carrier lifetime through PL analysis. However, the authors used other analysis e.g) TPD-MS, MPP stability, and device performance, comparing untreated perovskite to 4-ABPA ligand treated perovskite. If author's claim that DAA is a more important metric is correct, then the author should compare APSA treated perovskite and 4-ABPA treated perovskite device.
5. In this work, authors studied the degradation mechanisms of the wall in room temperature air, and moisture. Author performed MPP stability tracking of control and 4-ABPA treated perovskite solar cell in ambient condition in Figure 4i. However, in Page 13, these tests are performed using encapsulated devices. Would encapsulate device stability test fit into the study's proof that the ligand make the device more stable to air and moisture? In Figure 4i legend, the MPP test is conducted in continuous white LED illumination. Please specify the light source used for the MPP measurements.

Reviewer #2 (Remarks to the Author):

Sargent et al report a combined theoretical and experimental study on the dynamic behavior of perovskite surface defects with environmental stressors including heat, O₂, and H₂O. They found the dynamic surface adsorption affinity (DAA) relative to surface treatment ligands is a surrogate for passivation efficacy, then designed two new passivator molecules APSA and 4-ABPA based on FSA. Moreover, they conducted experimental measurements to validate their computational findings. The obtained results are important and the paper is well written. I recommend it publish in Nature Communications after addressing the following points.

1. Some AIMD details should be provided for clarity. For example, how many passivator molecules and O₂ molecules were used? Why? Where are the starting structures coming from?
2. There are some doubts about O₂ in AIMD simulations. Are the considered oxygen molecules neutral (O₂ in Fig.3d) or anionic (O₂⁻ in supplementary Fig.2)? In addition, it is well known the ground state of oxygen molecule is a triplet state. Was this point considered in this work? Why?
3. The quality of Fig. 3d needs to be improved. The different lines and arrows inside are not very clear.

Reviewer #3 (Remarks to the Author):

The authors claim a new method to determine the efficacy of passivating molecules for perovskite surfaces. They start with theoretical findings of defects on the various terminated surfaces that Pk can have using DFT combined with HSE and SOC. The changes of the defects are then analyzed under operating conditions, that is oxygen, moisture and heat. Based on the theoretical calculation the authors conclude that highly polar molecules are required to passivate the donor and acceptor defects on the Pk surfaces. The authors then investigate new molecules that have high and low polarity, similar binding energy to the PK surface and calculate the dynamic surface coverage of the different molecules under stressors like oxygen, moisture and heat. They find that mainly the surface coverage determines the passivation properties of the molecules and not necessarily the bind energy as previously thought. Then the most promising molecules are used to passivate the Pk surface experimentally. Various tests are performed in which 2 molecules (4-ABPA and APSA) are compared that have same binding energy but various surface coverage. The molecule that has theoretically the largest surface coverage (4-ABPA) also shows the higher stability (using TPD-MS), carrier lifetime and PL yield. When making devices only the most promising molecule (4-ABPA) is used and compared with a device without passivating molecule. Thus no comparison in device performance between the different investigated molecules (4-ABPA and APSA) is presented which would make the statement that the surface coverage can be used as a surrogate for passivation stronger. The efficiency of the low band gap Pk (Pb-Sn based) is about 21% and the T80 stability under ambient atmosphere (but encapsulated) is about 240 hours with the passivating molecules and 130 hours without the passivating molecules

There have been a few choices made that raised some questions. The DAA, dynamic adsorption affinity, which is introduced as the new surrogate to determine the passivating properties of a new molecules, is determined using an oxygen and moisture exposed Pk surface. Practically, Pk devices are prepared in protected and inert atmospheres and the passivating molecules are added under protected/inert atmosphere as well. Understanding the molecular adsorption of the molecules on degraded surface does not seem the most logical choice. These modules should cover the clean surfaces and then avoid any reactions with oxygen and moisture. Why is this choice made? Can it be assumed that the coverage won't change when the surface is clean (not exposed to oxygen and moisture?). And how would the coverage change when the passivated surface is exposed to moisture and oxygen? Another point about the DAA surrogate is that the concept is proven on new molecules. It would also be interesting to have a DAA of well-known molecules as well.

The passivation effect of the different molecules (4-ABPA and APSA) was seen in improved PL yield and carrier lifetime compared to an unpassivated surface. However, only the increase in yield is

given and not the increase in quasi-fermi-level-splitting. QFLS is required to have a fair comparison with improved Voc of the device. It seems that the PL yield has not increased to such an extent that it explains the Voc improvement in the devices (qfls \sim 20 mV, Voc \sim 60 mV). This difference needs to be investigated. Is it related to badge to badge reproducibility, the deposition of the top layer, or..? Also, it would have been good if devices were prepared using both molecules with similar binding energy but varying DAA (4-ABPA and APSA). It might then also be possible to relate the calculated surface coverage of the 2 molecules (4ABPA is about 11 sites of the 16 site available, for APSA this is about 8 sites) to the changes in Voc. This can then be transferred to reduction of expected defects and compared with device performance. This would make the conclusion that the DAA is a surrogate for surface passivation much more convincing.

In general, the stability improvements are quite low considering the cells were encapsulated and encapsulation should reduce the contact with oxygen and moisture to such an extent that mostly the intrinsic stability is being tested. Similar stability of 200 hours have been achieved before for Pb-Sn based perovskite before (under inert atmosphere).

Point-by-point list of author actions in response to Reviewer comments

Manuscript #: NCOMMS-23-37554-T

We thank the reviewers for their much-valued suggestions, which have enabled us to improve the manuscript. The following are detailed actions taken in light of reviewers' comments:

Referees' comments:

Reviewer #1 (Remarks to the Author):

In this study, the author introduced ab initio molecular dynamics (AIMD) simulation to reveal dynamic roles played by surface defects under atmospheric stressors and their chemical passivants. The author claimed the dynamic surface adsorption affinity (DAA) obtained by applying AIMD to perovskite is a surrogate for passivation efficacy. The ligand designed with consideration for the DAA (4-ABPA) improves the surface stability of perovskite. As a result, 4-ABPA treated perovskite enhanced a PCE of 21.6% and MPP stability over 240 hours under ambient conditions. This work involves applying interesting simulation techniques to confirm the surface stability of Sn corporate perovskite film. Surface stability is very important in perovskite; therefore, this work is important and novel. However, to make the study more solid, the following comments should be supplemented before publication.

A few comments:

1. In page 4, author claims that 'At the MAI-terminated perovskite surface, the dominant defect species which have a lower E_f are MA vacancies (VMA), iodine vacancies (VI), and iodine interstitials (Iinterstitial) situated at the outermost surface (LO) layer...'. The author should clarify where they found the lowest E_f in Figure 1d. For example, VPb has the lowest defect formation energy in most areas. But, VPb is not the dominant defect species.

Response: In the Methods section, we now explain that in defect calculations, the Fermi level will be pinned at a position corresponding to the crossing point of the lowest-energy donor-like and acceptor-like defects. We define the dominant defect species as the defects which have lower defect formation energy at this pinned fermi level position.

2. In page 5, author claims that 'the E_f of VMA in the bulk is 0.9 eV, while its value at the perovskite surface is reduced to 0.4 eV, causing a defect concentration that is eight orders of magnitude higher (at the surface compared to in the bulk) at room temperature. ...'. Is this finding a result of your work or previous research? The author should clarify the evidence for these statements.

Response: We now write in the revised manuscript to highlight that this is indeed a new finding from the present work:

“We found that, at the point B chemical potential condition and the pinned Fermi level (pinned E_{FL} , which is defined as the crossing point of the lowest-energy donor-like and acceptor-like defects¹⁷), the calculated E^f of V_{MA} in the bulk is 0.9 eV (Fig. 1c, middle panel), while its value at the perovskite surface is reduced to 0.4 eV (Fig. 1d, middle panel). This corresponds to a defect concentration that is (at room temperature) eight orders of magnitude higher at the surface compared to in the bulk, i.e. by a factor $e^{-\Delta E^f/k_B T}$ where k_B is the Boltzmann constant and T is the temperature.”

3. In page 7, the authors argued that AIMD simulation at RT(300K) can confirm O-O and Sn-I bonds breaking and formation of VSn at the surface. And this phenomenon were not observed in the static DFT calculation performed at 0K (Supplementary Fig 2). However, static DFT also can calculate the surface exposure at 300K. And in supplementary Fig 2b, we can find formation of VSn in static DFT simulation at 300K. Author should specifically and clearly describe the advantages of AIMD simulation (300K) over static DFT (300K). This should be a key part of this research.

Response: In the revised work we define “static DFT” as static first-principles calculations that exclude temperature effects. We better explain that they are not static calculations that follow an AIMD simulations at a finite temperature. We now write:

“These phenomena were not observed in static first-principles calculations that excluded temperature effects (Supplementary Fig. 2). This suggests the benefit that could be had in future studies wherein AIMD simulations are used to investigate temperature-dependent and time-dependent degradation process of perovskite materials; particularly for perovskite compositions which are highly sensitive to environmental conditions, such as mixed Pb-Sn perovskites.”

“Supplementary Fig. 2 Atomic structures of $MA_{0.25}FA_{0.75}Pb_{0.5}Sn_{0.5}I_3$ (001) surface exposure to O_2^- conditions (a) in static first-principles calculations that exclude temperature effects and (b) at 300K after ~ 10 ps AIMD simulations.”

4. In this work, the author argues that DAA relative to surface treatment ligand is a surrogate for passivation efficacy and DAA is a more strongly correlated feature than the static binding strength. The authors compared APSA and 4-ABPA ligands with similar static binding energies and showed that the 4-ABPA ligand with higher DAA had a better carrier lifetime through PL analysis. However, the authors used other analysis e.g) TPD-MS, MPP stability, and device performance, comparing untreated perovskite to 4-ABPA ligand treated perovskite. If author’s claim that DAA is a more important metric is correct, then the author should compare APSA treated perovskite and 4-ABPA treated perovskite device.

Response: For direct comparison with control and 4-ABPA-treated devices, we now provide the $J-V$ curves and device performance statistics data for APSA-treated Pb-Sn perovskite solar

cells (PSCs) in **Fig. 4g** and **Supplementary Fig. 10**, respectively. In this submission, we repeated the stability measurements and improved the way in which we encapsulated devices, resulting in better stability. We also present the MPP stability data for APSA-treated PSCs in **Fig. 4i**.

As shown in **Fig. 4g** and **Supplementary Fig. 10**, we observe a smaller improvement in PCE after APSA treatment (19.8 %) compared to 4-ABPA treatment (21.6 %). The PCE increases after each treatment relative to control devices are driven by improved device V_{OC} , and the overall trend is consistent with our PLQY and TRPL studies. This further suggests that while both ligands provide surface passivation, 4-ABPA is the more effective passivator overall.

We now include the below discussions in the revised manuscript:

“We also fabricated APSA-treated Pb-Sn devices, which delivered a V_{OC} and PCE of 0.83 V and 19.8%, respectively (**Fig. 4g**). The relative increase in V_{OC} is in good agreement with the trend observed in the PLQY measurements (**Fig. 4b**) and the calculated quasi-Fermi level splitting (QFLS) values (**Supplementary Note 3** and **Supplementary Fig. 11**).”

“The control device delivered a T_{80} (the time taken for device efficiency to drop to 80% of its initial value) of 130 hours. In comparison, the APSA treated device was reduced to 90% of its initial PCE after 170 hours, while the 4-ABPA treated Pb-Sn devices show little degradation after 290 hours. These findings suggest that the 4-ABPA treatment substantially enhances device stability over the control and APSA treatments.”

5. In this work, authors studied the degradation mechanisms of the cell in room temperature air, and moisture. Author performed MPP stability tracking of control and 4-ABPA treated perovskite solar cell in ambient condition in Figure 4i. However, in Page 13, these tests are performed using encapsulated devices. Would encapsulate device stability test fit into the study’s proof that the ligand make the device more stable to air and moisture? In Figure 4i legend, the MPP test is conducted in continuous white LED illumination. Please specify the light source used for the MPP measurements.

Response: We agree with the reviewer that ideally, device stability measurements should be carried out in ambient conditions. However, we note that as the perovskite composition used herein is that of mixed Pb-Sn, the rapid degradation due to Sn oxidation upon exposure to moisture and oxygen is a major factor. The PCE and PLQY of an unencapsulated ~ 1.2 eV Pb-Sn device will degrade within minutes when exposed to ambient conditions (V. J. Lim et al., *Adv. Energy Mater.* 2023, 13, 2200847), making it difficult to ascertain trends. Even trace amounts of moisture and oxygen, consistent with the levels within a N_2 -filled glovebox or encapsulated device (ie. 10 ppm for instance), can have a significant impact on device performance over the studied timescale. Based on these considerations, we now explain that improved stability against air and moisture is in fact better observed in MPP testing of encapsulated Pb-Sn perovskite solar cells.

Regarding the illumination used for MPP tracking, we used a broadband white LED with an

overall power density of approximately 1 sun (100 mW cm⁻²). We now provide the spectrum of the LED simulator we used in **Supplementary Fig. 16**.

Reviewer #2 (Remarks to the Author):

Sargent et al report a combined theoretical and experimental study on the dynamic behavior of perovskite surface defects with environmental stressors including heat, O₂, and H₂O. They found the dynamic surface adsorption affinity (DAA) relative to surface treatment ligands is a surrogate for passivation efficacy, then designed two new passivator molecules APSA and 4-ABPA based on FSA. Moreover, they conducted experimental measurements to validate their computational findings. The obtained results are important and the paper is well written. I recommend it publish in Nature Communications after addressing the following points.

1. Some AIMD details should be provided for clarity. For example, how many passivator molecules and O₂ molecules were used? Why? Where are the starting structures coming from?

Response: We now provide AIMD details as follows in the Supplementary Information.

“We constructed the FA_{0.75}MA_{0.25}Pb_{0.5}Sn_{0.5}I₃ (001) surface in the lateral dimension of 24.8×25.0 Å. We initially put 16 passivator molecules at the perovskite surface to represent complete surface coverage within the simulation unit (4×4 supercell). The H₂O layer in the FA_{0.75}MA_{0.25}Pb_{0.5}Sn_{0.5}I₃/H₂O interface model employed the experimental density of liquid water. We used the Packmol software¹⁸ (<https://m3g.github.io/packmol/>) to build the starting structure of H₂O layer.”

Based on the molar volume of gas at 1 atm pressure and the volume percent of oxygen in air, we estimated that the number of O₂ molecules in the simulation cell used in our calculations is small (less than 1 molecules). From this analysis, we opted to only use two O₂ molecules in our simulation cell.

2. There are some doubts about O₂ in AIMD simulations. Are the considered oxygen molecules neutral (O₂ in Fig.3d) or anionic (O₂⁻ in supplementary Fig.2)? In addition, it is well known the ground state of oxygen molecule is a triplet state. Was this point considered in this work? Why?

Response: Neutral oxygen molecule (O₂) and superoxide ion (O₂⁻) species are the two main forms of oxygen used in simulations (Run Long *et al.*, J. Am. Chem. Soc. 144, 5543-5551 (2022); Oleg V. Prezhdo *et al.*, J Am Chem Soc 142, 14664-14673 (2020).). Upon exposure to light, participation of one photoexcited electrons can convert the neutral oxygen molecule into superoxide. We performed the AIMD simulations of perovskite surface exposure to O₂⁻ (Supplementary Fig. 2) in the VASP software where the total number of atoms in the simulation box is around 350. To obtain an ionic pseudopotential for O₂⁻ in VASP, we use the configuration of 1s^{1.5}2s²2p^{4.5} in which “half” an electron from the 1s shell is put into the valence shell 2p for both the oxygen atoms.

We perform the AIMD simulations of perovskite surface exposure to O₂ and H₂O (Fig. 2c, Fig. 3c and Fig. 3d) in the CP2K software where there are around 2000 atoms in the simulation box. The simulation box is too large to perform in VASP software. We choose to use neutral O₂ in these simulations because the ionic pseudopotential for O₂⁻ is not available in the CP2K software.

3. The quality of Fig. 3d needs to be improved. The different lines and arrows inside are not very clear.

Response: We now write in the revised **Fig. 3d** legend:

“Fig. 3d The number of adsorbed ligands for APSA and 4-ABPA at the conditions shown in Fig. 3c from 4ps to 12ps during the AIMD simulation. Ligand number of 16 represents complete surface coverage within the simulation unit. Blue line: APSA adsorption case; Red line: 4-ABPA adsorption case. The average value of adsorbed ligands during the last 2ps of the AIMD simulation for APSA and 4-ABPA are 8 and 11, respectively.”

Reviewer #3 (Remarks to the Author):

The authors claim a new method to determine the efficacy of passivating molecules for perovskite surfaces. They start with theoretical findings of defects on the various terminated surfaces that Pk can have using DFT combined with HSE and SOC. The changes of the defects are then analyzed under operating conditions, that is oxygen, moisture and heat. Based on the theoretical calculation the authors conclude that highly polar molecules are required to passivate the donor and acceptors defects on the Pk surfaces. The authors then investigate new molecules that have high and low polarity, similar binding energy to the PK surface and calculate the dynamic surface coverage of the different molecules under stressors like oxygen, moisture and heat. They find that mainly the surface coverage determines the passivation properties of the molecules and not necessarily the bind energy as previously thought. Then the most promising molecules are used to passivate the Pk surface experimentally. Various tests are performed in which 2 molecules (4-ABPA and APSA) are compared that have same binding energy but various surface coverage. The molecule that has theoretically the largest surface coverage (4-ABPA) also shows the higher stability (using TPD-MS), carrier lifetime and PL yield. When making devices only the most promising molecule (4-ABPA) is used and compared with a device without passivating molecule. Thus no comparison in device performance between the different investigated molecules (4-ABPA and APSA) is presented which would make the statement that the surface coverage can be used as a surrogate for passivation stronger. The efficiency of the low band gap Pk (Pb-Sn based) is about 21% and the T80 stability under ambient atmosphere (but encapsulated) is about 240 hours with the passivating molecules and 130 hours without the passivating molecules

Response: For direct comparison with control and 4-ABPA-treated devices, we now provide the *J-V* curves and device performance statistics data for APSA-treated Pb-Sn perovskite solar cells (PSCs) in **Fig. 4g** and **Supplementary Fig. 10**, respectively. Additionally, we now present the MPP stability data for APSA-treated PSCs in **Fig. 4i**.

As shown in **Fig. 4g** and **Supplementary Fig. 10**, we observe a smaller improvement in PCE after APSA treatment (19.8 %) compared to 4-ABPA treatment (21.6 %). The PCE increases after each treatment relative to control devices are driven by improved device V_{OC} , and the overall trend is consistent with our PLQY and TRPL studies. This further suggests that while both ligands provide surface passivation, 4-ABPA is the more effective passivator overall.

We now include the below discussions in the revised manuscript:

“We also fabricated APSA-treated Pb-Sn devices, which delivered a V_{OC} and PCE of 0.83 V and 19.8%, respectively (**Fig. 4g**). The relative increase in V_{OC} is in good agreement with the trend observed in the PLQY measurements (**Fig. 4b**) and the calculated quasi-Fermi level splitting (QFLS) values (**Supplementary Note 3** and **Supplementary Fig. 11**).”

“The control device delivered a T_{80} (the time taken for device efficiency to drop to 80% of its initial value) of 130 hours. In comparison, the APSA treated device was reduced to 90% of its initial PCE after 170 hours, while the 4-ABPA treated Pb-Sn devices show little degradation after 290 hours. These findings suggest that the 4-ABPA treatment substantially enhances device stability over the control and APSA treatments.”

There have been a few choices made that raised some questions. The DAA, dynamic adsorption affinity, which is introduced as the new surrogate to determine the passivating properties of a new molecules, is determined using an oxygen and moisture exposed Pk surface. Practically, Pk devices are prepared in protected and inert atmospheres and the passivating molecules are added under protected/inert atmosphere as well. Understanding the molecular adsorption of the molecules on degraded surface does not seem the most logical choice. These modules should cover the clean surfaces and then avoid any reactions with oxygen and moisture. Why is this choice made? Can it be assumed that the coverage won't change when the surface is clean (not exposed to oxygen and moisture?). And how would the coverage change when the passivated surface is exposed to moisture and oxygen? Another point about the DAA surrogate is that the concept is proven on new molecules. It would also be interesting to have a DAA of well-known molecules as well.

Response: We clarify that in our AIMD simulations (**Fig. 3c** and **Fig. 3d**), the passivator molecules were first added to a pristine (clean) perovskite surface, and then adsorption was studied under the environmental stressors. We now have clarified this point in the revised manuscript.

The reason we chose to study DAA under moisture and oxygen exposure is that in practice it is nearly impossible to avoid trace amounts of oxygen and moisture exposure. While this is

not usually a significant issue for most perovskite compositions (e.g., pure Pb perovskites), narrow-bandgap Pb-Sn perovskites suitable for applications in all-perovskite multijunction devices are far more susceptible to even trace amounts of oxygen and moisture, and thus we believe that at this stage it is still important to consider stability under atmospheric stressors.

The passivation effect of the different molecules (4-ABPA and APSA) was seen in improved PL yield and carrier lifetime compared to an unpassivated surface. However, only the increase in yield is given and not the increase in quasi-fermi-level-splitting. QFLS is required to have a fair comparison with improved Voc of the device. It seems that the PL yield has not increased to such an extent that it explains the Voc improvement in the devices (qfls ~ 20 mV, Voc ~ 60 mV). This difference needs to be investigated. Is it related to badge to badge reproducibility, the deposition of the top layer, or..? Also, it would have been good if devices were prepared using both molecules with similar binding energy but varying DAA (4-ABPA and APSA). It might then also be possible to relate the calculated surface coverage of the 2 molecules (4ABPA is about 11 sites of the 16 site available, for APSA this is about 8 sites) to the changes in Voc. This can then be transferred to reduction of expected defects and compared with device performance. This would make the conclusion that the DAA is a surrogate for surface passivation much more convincing.

Response: We now include the device performance of both of the investigated molecules (4-ABPA and APSA) in the revised manuscript (**Fig. 4g** and **Supplementary Fig. 10**). In addition, we now discuss the cause for the discrepancy in the relative increases of QFLS and Voc in **Supplementary Note 3**:

“Quasi-Fermi level splitting (QFLS) is calculated by the PLQY values at various excitation light intensities:

$$QFLS = k_B T \times \ln(PLQY \times S \times J_G / J_{0,rad})$$

where S is the sun-equivalent excitation intensity, (set to 1 here as an equivalent 1-sun excitation density was used) J_G is the generated current density at 1 sun (taken from device J_{SC}) and $J_{0,rad}$ the radiative recombination current in the dark (taken from the dark current value from Shockley-Queisser limit). We calculated the QFLS of the control, APSA and 4-ABPA perovskite films from the PLQY data to be 0.89 V, 0.90 V and 0.91 V respectively.

To investigate the cause for the discrepancy in the relative increases of QFLS and Voc, we measured the PLQY of each of the films with a ~30 nm layer of electron transport layer (C₆₀) deposited atop the perovskite (**Supplementary Fig. 11**), since the perovskite/C₆₀ interface is known to be a prominent source of non-radiative recombination in *p-i-n* devices.²⁴ After depositing C₆₀, the PLQY (QFLS) drops to 0.22% (0.82 V), 0.42% (0.84 V) and 0.78% (0.86 V) for control, APSA and 4-ABPA-treated stacks, respectively. It is evident that these results are more closely aligned to the device Voc than the PLQY of the neat films lacking carrier transport layers. Additionally, our findings indicate that 4-ABPA not only serves as a more effective passivator

of the neat perovskite, but also results in a smaller relative drop in PLQY after deposition of C₆₀ than both the control film and the APSA-treated film, and therefore less interface recombination. We should note that mismatch of QFLS and V_{OC} is very common in perovskite solar cells, as discussed in J. Warby's paper.²⁵

In general, the stability improvements are quite low considering the cells were encapsulated and encapsulation should reduce the contact with oxygen and moisture to such an extent that mostly the intrinsic stability is being tested. Similar stability of 200 hours have been achieved before for Pb-Sn based perovskite before (under inert atmosphere).

Response: In our initial submission, we conducted the MPP stability measurement in a lab that had a limited ability to control oxygen and moisture levels (eg. < 5 – 10 ppm O₂, H₂O).

In the present revised submission, we carried out stability measurements in a new lab in which the ability to control the atmosphere in gloveboxes was improved. We also improved the way we encapsulated our devices, resulting in better stability.

We now write in the revised manuscript:

“The control device delivered a T_{80} (the time taken for device efficiency to drop to 80% of its initial value) of 130 hours. In comparison, the APSA treated device was reduced to 90% of its initial PCE after 170 hours, while the 4-ABPA treated Pb-Sn devices show little degradation after 290 hours. These findings suggest that the 4-ABPA treatment substantially enhances device stability over the control and APSA treatments.”

REVIEWERS' COMMENTS

Reviewer #1 (Remarks to the Author):

The manuscript is clearly revised and now suitable to be published in Nature Communications.

Reviewer #2 (Remarks to the Author):

In the revised manuscript, the authors have addressed all the comments raised by the reviewers. The present version is suitable for publication in Nature Communications.

Reviewer #3 (Remarks to the Author):

Thank you for the responds to the comments. The comments have been addressed thoroughly and manuscript looks good.

Point-by-point list of author actions in response to Reviewer comments

Manuscript #: NCOMMS-23-37554A

Referees' comments:

Reviewer #1 (Remarks to the Author):

The manuscript is clearly revised and now suitable to be published in Nature Communications.

Response: We thank the referee for a constructive peer review process.

Reviewer #2 (Remarks to the Author):

In the revised manuscript, the authors have addressed all the comments raised by the reviewers. The present version is suitable for publication in Nature Communications.

Response: We sincerely thank the referee for advice along the way.

Reviewer #3 (Remarks to the Author):

Thank you for the responds to the comments. The comments have been addressed thoroughly and manuscript looks good.

Response: We thank the referee for a constructive peer review process.